# Phage Therapy for Cardiac Implantable Electronic Devices and Vascular Grafts: A Targeted Literature Review

**DOI:** 10.3390/pathogens13050424

**Published:** 2024-05-17

**Authors:** Matteo Passerini, Francesco Petri, Gina A. Suh

**Affiliations:** 1Department of Pathophysiology and Transplantation, University of Milano, 20122 Milan, Italy; matteo.passerini1@gmail.com; 2Department of Infectious Disease, ASST FBF Sacco Milano, 20157 Milan, Italy; petri.francesco@mayo.edu; 3ESGNTA–ESCMID Study Group for Non-Traditional Antibacterials, 4051 Basel, Switzerland; 4Division of Public Health, Infectious Diseases and Occupational Medicine, Department of Medicine, Mayo Clinic, Rochester, 55905 MN, USA

**Keywords:** bacteriophage, phage therapy, phage, LVAD infections, cardiac device infections

## Abstract

Infections of cardiac implantable electronic devices (CIEDs) and vascular grafts are some of the most dreaded complications of these otherwise life-saving devices. Many of these infections are not responsive to conventional treatment, such as systemic antibiotics and surgical irrigation and debridement. Therefore, innovative strategies to prevent and manage these conditions are warranted. Among these, there is an increasing interest in phages as a therapeutical option. In this review, we aim to collect the available evidence for the clinical application of phage therapy for CIED and vascular graft infections through literature research. We found 17 studies for a total of 34 patients. Most of the indications were left ventricular assist device (LVAD) (n = 20) and vascular graft infections (n = 7). The bacteria most often encountered were *Staphylococcus aureus* (n = 18) and *Pseudomonas aeruginosa* (n = 16). Clinical improvements were observed in 21/34 (61.8%) patients, with microbiological eradication in 18/21 (85.7%) of them. In eight cases, an adverse event related to phage therapy was reported. Phage therapy is a promising option for difficult-to-treat CIED and vascular graft infections by means of an individualized approach. Clinical trials and expanded access programs for compassionate use are needed to further unveil the role of phage therapy in clinical application.

## 1. Introduction

The introduction of cardiac implantable electronic devices (CIEDs) and vascular grafts has indeed revolutionized the treatment landscape for various cardiovascular conditions. These advancements provide valuable therapeutic options for patients suffering from advanced heart failure [1], cardiac arrhythmias [2], valvular [3], and vascular dysfunctions [4]. Nevertheless, CIEDs are not devoid of complications, with infections being particularly formidable among them. This is owing to their substantial influence on morbidity, mortality, and the associated costs within the healthcare system. One of the main challenges in treating CIED and vascular graft infections is the presence of the biofilm matrix, which acts as a protective shield, hindering the efficacy of antimicrobial agents and host immune responses. Moreover, biofilm’s presence heightens the risk of systemic dissemination of pathogens, complicating treatment and necessitating more aggressive interventions [5,6]. Thus, CIED and vascular graft infections are often managed with the removal of some or all parts of the device, when possible, and often with long-term antimicrobial suppression with implant retention [7]. Despite important advances addressing the prevention, diagnosis, and management of these infections, in some clinical scenarios, the standard of care is not sufficient to obtain clinical cure and microbiological eradication. Therefore, innovative approaches, such as bacteriophages, are worthy of exploration.

Bacteriophages, or phages, are viruses that infect and replicate within bacterial cells. They consist of genetic material—either DNA or RNA—encased in a protein coat. Phages are highly specific to certain bacterial strains, targeting and infecting them with precision. The potential of phages in addressing difficult-to-treat infections, particularly biofilm-associated infections, stems from their unique ability to penetrate the protective matrix of biofilms [8]. Furthermore, phages exhibit the capacity to circumvent antibiotic resistance through unique and distinct mechanisms [9].

In recent years, CIED and vascular graft infections have presented a compelling avenue for investigating the effectiveness and safety of phage therapy. Some contemporary clinical cases posed significant challenges in treatment, stemming from issues such as antimicrobial resistance, constraints related to performing appropriate surgical procedures due to comorbidities or technical limitations, and the persistence of infections. Consequently, we conducted this targeted literature review to consolidate the existing clinical insights into phage therapy for managing CIED and vascular graft infections. The emphasis is placed on elucidating the challenges and opportunities inherent in this innovative therapeutic approach for addressing such infections. 

## 2. Methods

A PubMed search was performed on 2 December 2023, using the following terms: “bacteriophage” OR “phage therapy” AND “CIED” OR “cardiac implantable electronic device” OR “VAD” OR “LVAD” OR “ventricular assistant device” OR “prosthetic valve” OR “vascular graft” OR “pacemaker” OR “ICD” OR “implantable cardioverter defibrillator” (Figure 1). Results were limited to human studies. There was no restriction on language and date. We also searched the references of the included records and the online repository of the European Congress of Clinical Microbiology and Infectious Diseases (ECCMID) 2023 and IDweek 2023. Furthermore, we included results from references of the included papers, pre-print papers, and papers published during the writing process. In the case of records where the same case report was reported, we included the clinical information reported in the latest published paper. For each case, we extracted demographic (age and sex), microbiological (type of bacteria involved), and clinical data (devices infected, adverse events, clinical outcome, and microbiological eradication). We also reported information regarding the treatment course (concomitant surgical procedures or antimicrobial therapy, number and type of phages, route of phages administration, highest dose of phages). Screening was performed using Covidence, while data extraction was through Microsoft Excel 16.83.

## 3. Results

A total of 17 reports were included in this review for a total of 34 patients. Eight studies reported more than one clinical case [10,11,12,13,14,15,16,17]. The remaining nine studies reported the experience of a single patient [18,19,20,21,22,23,24,25,26]. The characteristics of the included patients are reported in Table 1 and Table 2. Four patients were described in two different studies.

### 3.1. Demographic, Clinical, and Microbiological Characteristics

Twenty-six out of thirty-four cases came from Europe or the United States. The mean age was 55.5 (SD 18.6) years, and most of the patients were male (28/31, 90.3%, of the available data). Most clinical indications were due to infections of a left ventricular assistant device (LVAD) (20/34, 58.8%), followed by vascular graft infections (7/34, 20.6%). The two most reported microorganisms were *S. aureus* (n = 18/38, 47.4%, of the total isolates) and *P. aeruginosa* (n = 16/38, 42.1%, of the total isolates).

### 3.2. Phage Therapy and Concomitant Treatment

Seven patients were treated with just one type of phage, while the others with multiple phages. We observed the following routes of phage administration: in twelve cases, the route was only systemic (intravenous infusion); in thirteen cases, the route of the administration was topical (direct application on the lesion, in situ through a catheter, intraoperative); the remaining nine patients received both applications. The duration of treatment was variable, ranging from one single day to 16 weeks. Eighteen out of twenty-nine patients with available data were treated also with a surgical procedure, while all the patients received concomitant antimicrobial therapy.

### 3.3. Efficacy and Safety

Regarding the reported safety data, three patients reported an increase in liver tests associated with nausea (n = 1) [13], fever (n = 1) [2,16], or no symptoms (n = 1) [14]. One patient suffered from fever, wheezing, and shortness of breath after the phage infusion [11,16]. All these four patients did not withdraw from the phage treatment since the symptoms were not judged to be severe and improved spontaneously. One patient, treated with in situ infusion of phage through a CT-guided needle, experienced an aortic perforation after the procedure [18]. One patient died due to heart failure after the start of the treatment [19]. Three patients experienced breakthrough bacteremia [11,16]. Twenty-one out of thirty-four cases (61.8%) improved clinically. Among these, the bacteria were eradicated in 18/21 (85.7%) patients.

## 4. Discussion

The current review included 17 studies covering 34 patients with CIED and vascular graft infection treated with phage therapy. Our findings align with a recent study, which investigated the safety and efficacy of phage therapy in cardiac and peripheral vascular surgery by means of a systematic review encompassing 14 reports and 40 patients [27]. All 14 reports but one [28] were also included in our targeted literature review, which expands upon this with the inclusion of four additional studies [16,17,20,21], given the more recent literature review and the inclusion of pre-print papers. The only study not included in our review is a report of 550 cases, among whom seven developed pyopericardium after heart surgery [28]. However, we could not extract individual data, and we were not able to discern if a CIED or vascular graft had been used for these patients. Our investigation is a targeted literature review with the primary aim of addressing some questions arising from initial experiences with phage therapy used for CIED or vascular graft infections. Consequently, we also incorporate preliminary findings from notable studies, which we deemed capable of offering valuable insights given the researchers’ expertise in this domain. By contrast, the work by Simpsonet et al. constitutes a systematic review that extensively searches multiple databases with the aim to “review the evidence base for safety and efficacy” of the use of phage therapy in cardiovascular surgery [27].

In our study, phage administration was heterogeneous. The proportion of clinical and microbiological cures was significant; however, the limited sample size precludes drawing definitive conclusions in this regard. Nevertheless, the data obtained may help to address some common questions regarding the clinical use of phage therapy. It is important to acknowledge the potential for publication bias in our review. While we included 17 studies covering 34 patients with CIED and vascular graft infections treated with phage therapy, it is possible that cases where phage therapy failed may not have been reported. Thus, the findings presented here should be interpreted in light of this potential bias, recognizing the need for further investigation and reporting of both successful and unsuccessful outcomes in phage therapy for such infections.

### 4.1. Which Patients with CIED and Vascular Graft Infections Were Treated?

While clinical trials are ongoing, phage therapy is available through compassionate use programs in many countries. Therefore, the clinical choice for an intervention with an unproven efficacy should balance risks and benefits. Therefore, national regulatory frameworks governing the use of unproven (or unlicensed) interventions on a compassionate basis should be followed.

In this review, we found that most of the clinical indications were LVAD infections, which are usually affected by higher mortality [29] compared to other CIED infections [30], and prosthetic vascular graft infections, which can be challenging to treat given the high risk of surgical re-intervention [31]. Moreover, the two most common bacteria treated were *S. aureus* and *P. aeruginosa*. Although these bacteria are not the most frequent causes of such infections [7], they often cause difficult-to-treat infections due to biofilm formation [32,33] and other mechanisms of resistance [34]. In addition, some biobanks store phage targeting especially difficult-to-treat microorganisms such as *S. aureus* and *P. aeruginosa* [35]. The availability of already stored phage strains with a proven in vitro lytic activity may help save time for acute conditions such as some of the CIED and vascular graft infections.

Hence, challenging cases of difficult-to-treat infections associated with CIED and vascular grafts, particularly those caused by pathogens such as *S. aureus* and *P. aeruginosa*, represent potential candidates for phage therapy.

### 4.2. How Were Phages Administered?

Regarding the route of phage administration, our review found a heterogeneous approach. However, this should not be interpreted only as a limitation. Due to the narrow host range, phage therapy is, per se, an individualized treatment, and the possibility to administer phages in different preparations [36] enables a more tailored approach, especially for compassionate use. Twenty-three patients received the phages directly into the infected site, by direct application, through a catheter, or in the operating room; this may enable phages to easily reach the infected device and replicate in the bacterial biofilm [37].

### 4.3. What Is the Role of Concomitant Antibiotics?

All the patients in this review were also treated with conventional antibiotics. According to current knowledge, phage therapy for CIED and vascular graft infections may be an adjunct to conventional treatment rather than an alternative. Two noteworthy insights from the included studies merit attention. For a broader discussion on interactions between phage and antibiotics, we refer to a more comprehensive work [38].

First, Green and colleagues employed a methodology termed “synography” to discern the optimal phage–antibiotic combination [15], drawing from prior findings suggesting the potential for synergy, antagonism, or additive effects between phage and antibiotics [39]. The essence of the “synogram” entails subjecting isolated bacteria to escalating concentrations of phage and diverse antibiotics, followed by quantifying bacterial concentration reduction at distinct time intervals via a heatmap visualization. This method enables researchers to visually evaluate the impact of combining varying concentrations of phage and antibiotics on bacterial growth dynamics over time. Through analysis of the resulting heat map derived from synography, researchers can discern synergistic, antagonistic, or additive interactions between the phage and antibiotics, facilitating the selection of optimal combinations for clinical application in bacterial infection treatment. Second, Blasco and colleagues described a case of vascular graft infection due to multidrug-resistant *P. aeruginosa* [25]. After the application of the phage treatment, a new *P. aeruginosa* infection occurred, but involving a wild-type strain being susceptible to ß-lactams and quinolones. This resensitization was postulated to be due to phage action, which downregulated membrane receptors, regulators, and efflux pumps involved with antibiotic resistance.

These two findings suggest that the interaction between phages and antibiotics is complex and may not be limited to a simple additive effect of phages and antibiotic therapy. Given the complexity inherent in phage/antibiotic interactions, their efficacy is likely to vary by dose, specific phage, and antibiotic used. Further investigation into these interactions is crucial for optimizing treatment strategies in complex infections.

### 4.4. What Is the Role of Surgery?

Surgery is one of the mainstays of the management of CIED and vascular graft infections. In our review, 18/34 (52.9%) patients underwent surgery. All these patients also received topical administration of the phage therapy directly in the operating room or through an in situ catheter to allow administration in the days following the procedures. Therefore, surgery can first provide a therapeutic option for the patients with an indication for surgical procedures. Second, it can also provide a way to administer phages directly into the infected site, potentially increasing the chance of microbiological eradication and preventing the colonization of new devices inserted. An optimal example of this dual role of surgery can be found in a study [23] where the physicians performed a three-step approach for both extra- and endovascular application of phage therapy to treat a complex aortic stent graft infection.

### 4.5. Is There a Pattern between Management and Outcomes?

Regarding efficacy and safety, determining the true effect of this innovative therapy is challenging at the present state due to the small number of patients included and the high risk of confounders. Nevertheless, it is interesting to present the data obtained. Among the patients undergoing surgery, clinical improvement was observed in 15/18 (83.3%) versus patients without surgery in 6/16 (37.5%). This suggests the great impact of surgery on the clinical cure, but also the potential role of phage in patients not deemed candidates for surgery. Overall, the proportion of clinical improvement (21/34, 61.8%) was considerable, given that all the patients failed conventional treatment. Importantly, 18/21 (85.7%) patients with clinical improvement also showed microbiological eradication.

### 4.6. Is Phage Therapy Safe?

Eight patients reported adverse events. Two severe adverse events were related to the surgical procedure or concomitant heart failure rather than the phage administration, although in the latter, endotoxin release could not be excluded as a contributing factor to the patient’s decompensation 36 h after initial phage administration [19]. Whether there was causality between the reported adverse events and the administration of phage therapy is difficult to discern, given the lack of large-scale data. However, this cannot be ruled out, especially considering the time correlation. As suggested by Aslam et al., even if the source of the fever was unclear, it was plausible that pyrogens could have been present in the bacteriophage solution, potentially linked to solvents utilized during dilution or manufacturing processes [11]. These pyrogens might have been subsequently diluted with lower concentrations, leading to the absence of further adverse events in the patient. In the study by Tkhilaishvili et al., the laboratory values normalized following a reduction in intravenous bacteriophage dosage [13]. Consequently, the authors found it challenging to ascertain whether continuing intravenous phage therapy at the same dosage would have worsened liver function or resulted in adaptation and resolution. These two cases may potentially indicate a dose-dependent increase in adverse events; however, drawing conclusions from such limited experience is unfounded. These two cases might possibly suggest a dose-dependent increase in adverse events, but no conclusions can be drawn from such limited experience. In addition, the interplay between the immune system of the host and the phage preparation, as well as the predictability and the cause–effect relationship of such adverse events, still need further research. However, high-quality manufacturing is paramount for the prevention of future adverse events. Three patients reported breakthrough bacteremia during the phage treatment. All these patients were affected by LVAD infections due to *P. aeruginosa*. The authors proposed that the phenomenon might stem from pathogen release into the bloodstream due to the rapid degradation of the device biofilm by phages, given the susceptibility of the isolates to the administered phages. Alternatively, they suggested that the phages might swiftly eliminate certain isolates, creating space for less susceptible ones to dominate the ecological niche [11,16].

## 5. Conclusions

Phage therapy is a promising complementary tool against CIED and vascular graft infections. As new data are emerging and while clinical trials are conducted, some patients can be considered for phage therapy via compassionate use. This selection should especially consider patients with recurrent infections due to bacteria resistant to conventional antibiotics and potentially amenable to surgical debridement. Attention to the potential emergence of adverse events, including unexpected ones, and the investigation of potential failure factors are warranted.

## Figures and Tables

**Figure 1 pathogens-13-00424-f001:**
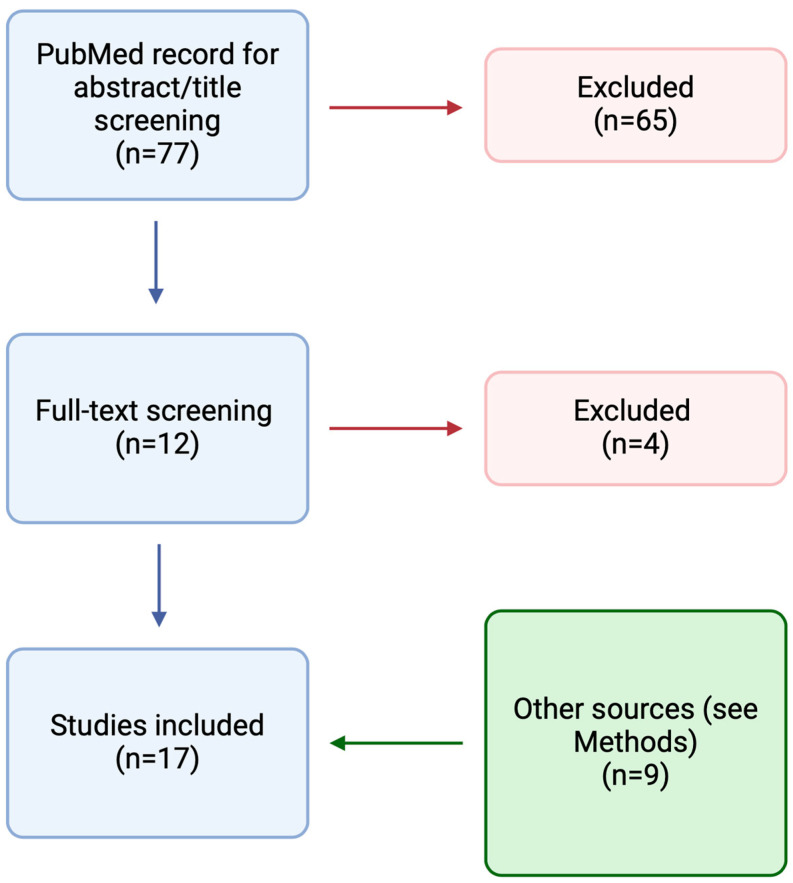
Results of the PubMed search resulting in the inclusion of 17 studies.

**Table 1 pathogens-13-00424-t001:** Demographic, microbiological, and clinical characteristics of the 34 patients included, combined with the features of the phage treatment.

Patient	Article Reference	Year	Country	Age	Sex	Pathogen	Type Infection	Number/Type of Phage(s)	Route of Phage Administration	Highest Dose of Phage Administered (PFU/mL)	Duration
**1**	**Chan, BK** [18]	2018	US	76	M	*P. aeruginosa*	Vascular graft infection	One (OMKO1)	In situ through a CT-guided needle	1 × 10^7^	Once
**2**	**Duplessis, C** [19]	2018	US	2	M	*P. aeruginosa*	bacteremia after the ASD/VSD closures	Two (not specified)	iv	3.5 × 10^5^	q6h for six doses, resumed after 11 days
**3**	**Rubalskij, E (1)** [10]	2020	Germany	52	M	*S. aureus* *E. faecium* *P. aeruginosa*	Vascular graft infection	Four (CH1, Enf1, PA5, PA10)	Topically + iv + intraoperative	1 × 10^8^	q24h for 2 separate days
**4**	**Rubalskij, E (2)** [10]	2020	Germany	59	M	*S. aureus*	Vascular graft infection	One (CH1)	In situ through a chest tube	1 × 10^9^	q12h for 2 days
**5**	**Rubalskij, E (3)** [10]	2020	Germany	62	M	*S. aureus*	Pleural empyema after LVAD implantation	One (CH1)	In situ through a drainage	1 × 10^9^	q12h (14 doses)
**6**	**Rubalskij, E (4)** [10]	2020	Germany	51	M	*S. aureus*	LVAD	Four (Sa30, CH1, SCH1, SCH111)	In situ through a catheter + intranasal + per os	1 × 10^9^	9 + 6 days
**7**	**Rubalskij, E (5)** [10]	2020	Germany	45	M	*S. aureus*	Implantable infusion pump	One (Sa30)	Intraoperative (new pump covered)	4 × 10^10^	Once
**8**	**Rubalskij, E (6)** [10]	2020	Germany	66	M	*E. coli*	Sternal wall healing disorder after mitral valve replacement and AoCo bypass surgery	Two (ECD7, V18)	Intraoperative	5 × 10^10^	Once
**9**	**Aslam, S (1)** [11]	2020	US	65	M	*S. aureus*	LVAD	Three (AB-SA01)	iv	3 × 10^9^	q12 for 28 days
**10**	**Aslam, S (2)** [11]	2020	US	64	M	*P. aeruginosa*	Vascular graft infection	Three (PPM2)	iv	2.6 × 10^6^	q12h for 6 weeks
**11**	**Mulzer, J** [20]	2020	Germany	67	M	*S. aureus*	LVAD	Two (PYO, Sb-1)	In situ through a catheter	1 × 10^7^	q8h for 10 days
**12**	**Gilbey, T** [21] **and Petrovic Fabijan, A (1)** [12]	2019 and 2020	Australia	65	M	*S. aureus*	Prosthetic valve endocarditis	Three (AB-SA01)	iv	1 × 10^9^	q12h for 14 days
**13**	**Petrovic Fabijan, A (2)** [12]	2020	Australia	69	F	*S. aureus*	Prosthetic valve endocarditis	Three (AB-SA01)	iv	1 × 10^9^	q12h for 14 days
**14**	**Petrovic Fabijan, A (3)** [12]	2020	Australia	21	M	*S. aureus*	Prosthetic valve endocarditis	Three (AB-SA01)	iv	1 × 10^9^	q12h for 28 days
**15**	**Petrovic Fabijan, A (4)** [12]	2020	Australia	81	M	*S. aureus*	Prosthetic valve endocarditis	Three (AB-SA01)	iv	1 × 10^9^	q12h for 14 days
**16**	**Puschel, A** [22]	2022	Germany	57	M	*P.mirabilis* *S. aureus*	LVAD	Cocktail (SniPha 360)	In situ in the interface between driveline and coating	1 × 10^7^	Once
**17**	**Grambow, E** [23]	2022	Germany	67	F	*S. aureus*	Vascular graft infection	Cocktail (SniPha 360)	Intraoperative + in situ through VAC system + hydrogel at the second stage of the surgery + stent graft covered with hydrogel	NA	6 days
**18**	**Rojas, SV** [24]	2022	Germany	49	M	*S. aureus*	LVAD	Cocktail (SniPha 360)	Intraoperative	1 × 10^7^	Once
**19**	**Tkhilaishvili, T (1)** [13]	2022	Germany	41	M	*S. aureus* *C.acnes*	LVAD + vascular graft infection	Two (PYO, Sb-1)	In situ through a drainage + intraoperative	1 × 10^7^	q8h for 14 days
**20**	**Tkhilaishvili, T (2)** [13]	2022	Germany	67	M	*S. aureus*	Abscess at the thoracotomy scar site with connection to the LVAD pump	Two (Sb-1, Pyo)	In situ through a drainage + intraoperative	1 × 10^7^	q8h, 10 days
**21**	**Tkhilaishvili, T (3)** [13]	2022	Germany	68	M	*P. aeruginosa*	LVAD	Two (Autophage [personalized phage], PYO)	Topically	1 × 10^8^	q12h for 12 days (Autophage), q12h for 5 days (PYO)
**22**	**Tkhilaishvili, T (4)** [13]	2022	Germany	53	M	*P. aeruginosa*	LVAD	Three (PNM, 14/1, PT07)	Topically + iv + intraoperative	1 × 10^8^	q12h for 5 days topical + once iv
**23**	**Onallah, H (1)** [14]	2023	NA	29	M	*P. aeruginosa*	LVAD	Three (PASA16, SaWIQ0488Phi, PaWRA02Phi87)	Topically + iv + intraoperative	4.7 × 10^10^	q24h for 2 weeks
**24**	**Blasco, L** [25]	2023	Spain	50	M	*P. aeruginosa*	Vascular graft infection	Three (PNM, 14/1, PT07)	iv	2.1 × 10^9^	q24h for 7 days
**25**	**Racenis, K** [26]	2023	Latvia	50	M	*P. aeruginosa*	LVAD	Two (PNM, PT07)	In situ through a catheter + iv	1 × 10^7^	q24h for 8 days (iv) + q24h for 4 days (topical)
**26**	**Green, SI** [15]	2023	US	67	M	*S. aureus*	LVAD	Two (SA4, DUD2)	iv + intraoperative	3 × 10^10^	q12h for 6 weeks
**27**	**Green, SI** [15]	2023	US	73	M	*P. aeruginosa*	LVAD	Two (6917, 6959)	iv + in situ	1 × 10^11^	q12h for 6 weeks
**28**	**Green, SI** [15]	2023	NA	NA	NA	*S. aureus*	LVAD	One (SA4)	iv	1 × 10^9^	q12h for 6 weeks
**29**	**Aslam, S (3)** [11] **and Aslam, S (1)** [16]	2020 and 2024	US	60	M	*P. aeruginosa*	LVAD	Three (GD-1)	iv	1.9 × 10^7^	q8h for 6 weeks
**30**	**Aslam, S (4)** [11] **and Aslam, S (2)** [16]	2020 and 2024	US	82	M	*P. aeruginosa*	LVAD	*First treatment* (i) initially two (PAK_P1, E127) (ii) then one (PAK_1) (iii) then two (PAK_P1, PAK_P5) *Second treatment* (iv)four (PPM3)	*First treatment* topically + iv *Second treatment* iv	4 × 10^10^	*First treatment* (i) q8h for 2 weeks (ii) q4h for 11 weeks (iii) q12h for 3 weeks *Second treatment* (iv) q12h for 6 weeks
**31**	**Onallah, H (2)** [14], **Aslam, S (3)** [16]	2023 and 2024	Israel	10	F	*P. aeruginosa*	LVAD	One (PASA16)	iv	1.72 × 10^11^	q12h for 49 days
**32**	**Aslam, S (4)** [16]	2024	Israel	52	M	*P. aeruginosa*	LVAD	One (PASA16)	iv	5 × 10^10^	q24h for 2 weeks
**33**	**Pirnay, JP (1)** [17]	pre-print	Belgium	NA	NA	*P. aeruginosa*	LVAD	Three (14-1, PNM, PT07)	Intraoperative + iv + topical through a dressing	1 × 10^8^	q24h for 10 days (iv) + q12h for 5 days (topical)
**34**	**Pirnay, JP (2)** [17]	pre-print	Belgium	NA	NA	*P. aeruginosa*	LVAD	Two (PNM, PT07)	iv + intralesional	1 × 10^7^	q24h for 8 days (iv) + q24 for 5 days (intralesional)

**Acronyms:** PFU: plaque-forming unit; CT: computed tomography; iv: intravenously; LVAD: left ventricular assistant device; AoCo: aortocoronary; VAC: vacuum-assisted closure; NA: not available; ASD: atrial septal defect; VSD: ventricular septal defect.

**Table 2 pathogens-13-00424-t002:** Clinical and microbiological outcomes.

Patient	Article Reference	Concomitant Surgery	Concomitant Antibiotics	Adverse Events (Type)	Clinical Improvement	Eradication of the Targeted Bacteria	Follow-Up	On Antimicrobial Suppression at Follow-Up
**1**	**Chan, BK** [18]	No	Yes	Yes (aortic perforation due to the CT-guided procedure)	No	Yes	18 months	No
**2**	**Duplessis, C** [19]	No	Yes	Yes (heart failure)	No	Yes	Patient died during treatment course	NA
**3**	**Rubalskij, E (1)** [10]	Yes	Yes	No	No	Yes	2 months, patient died due to a new infection	NA
**4**	**Rubalskij, E (2)** [10]	No	Yes	No	Yes	Yes	7 months after bacteriophage therapy, no signs of infection at PET/CT	NA
**5**	**Rubalskij, E (3)** [10]	Yes	Yes	No	Yes	Yes	20 months. Patient died after heart transplant due to transplant failure	NA
**6**	**Rubalskij, E (4)** [10]	No	Yes	No	No	No	Patient died 1.5 months after beginning phage therapy due to *S. aureus* sepsis	NA
**7**	**Rubalskij, E (5)** [10]	Yes	Yes	No	Yes	Yes	1.5 months after phage therapy, no signs of infection were observed	NA
**8**	**Rubalskij, E (6)** [10]	Yes	Yes	No	Yes	Yes	Alive at last follow-up	NA
**9**	**Aslam, S (1)** [11]	No	Yes	No	Yes	No	7 months	No
**10**	**Aslam, S (2)** [11]	No	Yes	No	Yes	Yes	3 months	NA
**11**	**Mulzer, J** [20]	Yes	Yes	No	Yes	Yes	9 months after discharge, no local signs of infection	NA
**12**	**Gilbey, T** [21] **and Petrovic Fabijan, A (1)** [12]	No	Yes	No	No	Yes	3 months	No
**13**	**Petrovic Fabijan, A (2)** [12]	No	Yes	No	Yes	Yes	3 months (additional surgery day 50, death day 90)	NA
**14**	**Petrovic Fabijan, A (3)** [12]	Yes	Yes	No	No	No	3 months (additional surgery day 15 followed by a subsequent phage cycle)	NA
**15**	**Petrovic Fabijan, A (4)** [12]	No	Yes	No	No	NA	Patient died day 27 of respiratory failure	NA
**16**	**Puschel, A** [22]	Yes	Yes	No	Yes	No	2 months after discharge, patient readmitted for mild local infection at the driveline exit by *S. aureus* only	No
**17**	**Grambow, E** [23]	Yes	Yes	No	Yes	Yes	12 months	No
**18**	**Rojas, SV** [24]	Yes	Yes	No	Yes	No	3 months, PET/CT with significant decrease of the outflow graft infection. Patient readmitted 6 months later with local infection of the driveline exit with *S. aureus* without interrelation to the former focus	No
**19**	**Tkhilaishvili, T (1)** [13]	Yes	Yes	No	Yes	Yes	12 months	No
**20**	**Tkhilaishvili, T (2)** [13]	Yes	Yes	No	Yes	Yes	30 months	No
**21**	**Tkhilaishvili, T (3)** [13]	Yes	Yes	No	Yes	Yes	Relapsed	NA
**22**	**Tkhilaishvili, T (4)** [13]	Yes	Yes	Yes (slight increase in GGT and direct bilirubin, mild nausea)	Yes	Yes	Patient died 4 months after bacteriophage therapy due to LVAD pump thrombosis, no signs of infection	No
**23**	**Onallah, H (1)** [14]	No	Yes	Yes (elevated ALK)	Yes	Yes	3 months	No
**24**	**Blasco, L** [25]	No	Yes	No	No	No	10 months	No
**25**	**Racenis, K** [26]	Yes	Yes	No	Yes	Yes	6 weeks after driveline repositioning, slight residual metabolic activity at PET/CT; no signs of inflammation at the wound. PET/CT negative at 34 months. No clinical recurrence at 21 months	No
**26**	**Green, SI (1)** [15]	Yes	Yes	No	Yes	Yes	32 months	No
**27**	**Green, SI (2)** [15]	Yes	Yes	No	Yes	Yes	27 months	No
**28**	**Green, SI (3)** [15]	NA	Yes	No	No	No	Bacteremia recurred after the end of phage and antibiotic therapy	Yes
**29**	**Aslam, S (3)** [11] **and Aslam, S (1)** [16]	No	Yes	No	No	No	7 months (when it was performed a heart transplant with surgical cultures positive for *P. aeruginosa*)	Yes
**30**	**Aslam, S (4)** [11] **and Aslam, S (2)** [16]	Yes	Yes	Yes (fever, wheezing, and shortness of breath)	No	No	4.5 months (when the patient died due to a septic event)	Yes
**31**	**Onallah, H (2)** [14] **and Aslam, S (3)** [16]	No	Yes	Yes (elevated LFT 2xULN, high fever)	No	No	2 months (when the patient died)	No
**32**	**Aslam, S (4)** [16]	No	Yes	No	No	No	0.5 months	Yes
**33**	**Pirnay, JP (1)** [17]	Yes	Yes	No	Yes	Yes	NA	NA
**34**	**Pirnay, JP (2)** [17]	No	Yes	No	Yes	Yes	NA	NA

**Acronyms:** CT: computed tomography; NA: not available; PET: positron emission tomography; GGT: gamma glutamyl transferase; LVAD: left ventricular assistant device; ALK: alkaline phosphatase; LFT: liver function tests; ULN: upper limit normal.

## Data Availability

No new data were created or analyzed in this study. Data sharing is not applicable to this article.

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
