# Peer review of "Phage Therapy for Cardiac Implantable Electronic Devices and Vascular Grafts: A Targeted Literature Review"

_pathogens, 2024, doi:10.3390/pathogens13050424_

Round 1

Reviewer 1 Report

Comments and Suggestions for Authors

This manuscript seeks to review the applications of phage therapy cardiac implantable electronic device and vascular graft infections.

Major issues

1.      The nature of the review is unclear. This review is not presented as a systematic review but attempts to employ such methods. However, if viewed as a systematic review, there are several major shortcomings. These include:

a.      Use of only one database and no justification for the sources chosen for hand-searching.

b.      No presentation of reasons why papers were excluded at the full text stage.

c.      No PRISMA statement.

d.      There is no discussion of review limitations.

2.      This manuscript should be seen in the context of a similar paper, published in the MDPI journal Antibiotics entitled: The Safety and Efficacy of Phage Therapy for Infections in Cardiac and Peripheral Vascular Surgery: A Systematic Review. This draft manuscript should contain a discussion of its context relative to the closely related published piece, including addressing why it identifies fewer patients.

3.      It is an inappropriate generalisation to suggest the Declaration of Helsinki is appropriate in all contexts. Instead, it would be more appropriate to say that national regulatory frameworks governing the use of unproven (or unlicensed) interventions on a compassionate basis should be followed.

4.      The section about selection of patients for phage therapy is descriptive of the nature of the patients that were selected, not how to select; please re-position how this section is presented to reflect this. Notwithstanding the previous major issue, your comment regarding compassionate use adequately covers patient selection. So small as the number of patients in this cohort that it is not possible, nor appropriate given the unique and varied nature of suitable cases, to attempt to make any more precise comments regarding selection.

5.      Similar to the previous major issue, so small is the cohort of patients reviewed that it is not possible or appropriate to present methodology about the administration of phage in a manner that could be read as advisory: e.g. ‘how to administer phages’. Instead, I suggest presenting that information as ‘experience of phage administration’ or ‘how phages were administered’.

6.      Line 149/150 – ‘intravenous is not easy due to regulatory issues’ – this statement is not justified and should be removed. I am not aware in my, or any other, context of any additional regulatory barriers to IV vs. local phage use.

7.      Line 157 – ‘is not an alternative’ – another generalisation. Phage therapy may be an alternative in some circumstances, e.g. allergy to antibiotics: https://www.ncbi.nlm.nih.gov/pmc/articles/PMC5494523 

8.      The discussion about phage/antibiotic interactions needs to be more succinct and precise. ‘As showed in some studies in this review’ is followed by no references. The discussion about phage/antibiotic interactions is not placed sufficiently in the context of broader evidence in this area.

9.      ‘Is there any pattern between management and outcomes?’ This section doesn’t really answer this question. Line 189 suggests answering this question is difficult due to the high risk of confounding factors, but these are not explored. There is no mention of this being challenging due to be extremely small number of patients considered in this review.

10.  The safety of phage therapy is such an important aspect of the evidence base that it requires greater consideration, both in scope and quality, than has been given. Line 200 speculates about all reported adverse events being related to the encounter between the phage preparation and the human immune system. This speculatory statement is not justified and should be amended or removed. Moreover, alternative and more plausible explanations are presented by the authors in the original reports of these adverse events. No detail is given on what a ‘more standardised approach for the preparation of the phage treatment’ means - presumably manufacturing in accordance with good manufacturing practice?

11.  It is unclear what the value the section about workflow adds. Suggest this is removed. The workflows will be different in different clinical settings and in different national contexts. As presented, the authors note that ‘most’ but not all cases were treated in specialised centres. The authors should avoid suggesting that only specialised centres are able to deliver phage therapy, as this is not the case as they have acknowledged. The authors also imply that the effort and regulatory requirements necessitate a specialised centre. It is incorrect to suggest that other centres are not capable of equally significant effort or are incapable of following the same regulations. It is also generalisation to state that collecting data and demonstrating the connectivity are often mandatory, when again the requirements here will vary markedly between national contexts.

Minor issues

1.      There are more appropriate references regarding anti-biofilm activity than reference 8.

2.      Rephrase line 48/49 – phages don’t ‘overcome’ antibiotic resistance, as it’s not a barrier phages need surmount.

3.      Software versions should be stated.

4.      Line 87 – be consistent, use n=X/29 etc to orientate the reader.

5.      Line 89 – ‘combination of multiple’ is tautologous.

6.      Please reference the reports you refer to when discussing efficacy/safety.

7.      Line 1030-104 typo ‘wewewe’ and duplication.

8.      Ensure bacterial genus/species are presented consistently. I.e. in full first and then abbreviated, e.g. S. aureus. See line 138.

9.      Line 143/144 does not read well.

10.  Fig 2 has no labels; the legend suggests there should be.

11.  There appear to be some instances of double spaces.

Comments on the Quality of English Language

Requires moderate English editing. See comments for examples of areas requiring attention.

Author Response

Response to Reviewer 1 Comments

We thank the reviewer for the thoughtful comments and offer our responses below. We feel that the manuscript is stronger as a result of the suggestions.

Major issues

1.      The nature of the review is unclear. This review is not presented as a systematic review but attempts to employ such methods. However, if viewed as a systematic review, there are several major shortcomings. These include:

a.      Use of only one database and no justification for the sources chosen for hand-searching.

b.      No presentation of reasons why papers were excluded at the full text stage.

c.      No PRISMA statement.

d.      There is no discussion of review limitations.

Response: The reviewer is correct in that the nature of the review is not systematic. We did not intend to perform a systematic review following the PRISMA indications, since, as kindly mentioned by the reviewer, another systematic review on a similar topic has already been performed. Rather, we collected recent papers and highlighted learning points that may help the treating physician in clinical practice. This approach was recently used in another high-quality journal (https://doi.org/10.1093/cid/ciad533). We have changed the title to better reflect this comment on methodology to “Phage Therapy For Cardiac Implantable Electronic Devices and Vascular Grafts: A Targeted Literature Review”). (A reference on targeted systematic reviews: https://www.ncbi.nlm.nih.gov/books/NBK553274/).

2.      This manuscript should be seen in the context of a similar paper, published in the MDPI journal Antibiotics entitled: The Safety and Efficacy of Phage Therapy for Infections in Cardiac and Peripheral Vascular Surgery: A Systematic Review. This draft manuscript should contain a discussion of its context relative to the closely related published piece, including addressing why it identifies fewer patients.
Response: We thank the reviewer for this suggestion. We read with interest the suggested paper and we inserted in the Discussion section some sentences to comment the similarities and the differences with that study. Moreover, we checked the references of this systematic review and we found 3 additional studies to include. We added these 3 studies in our paper (please see Table 1 and Table 2) and we also added a recently published paper by Aslam et al.

3.      It is an inappropriate generalization to suggest the Declaration of Helsinki is appropriate in all contexts. Instead, it would be more appropriate to say that national regulatory frameworks governing the use of unproven (or unlicensed) interventions on a compassionate basis should be followed.

Response: We modified the text accordingly.

4.      The section about selection of patients for phage therapy is descriptive of the nature of the patients that were selected, not how to select; please re-position how this section is presented to reflect this. Notwithstanding the previous major issue, your comment regarding compassionate use adequately covers patient selection. So small as the number of patients in this cohort that it is not possible, nor appropriate given the unique and varied nature of suitable cases, to attempt to make any more precise comments regarding selection.
Response: We have rephrased the title to reflect the comment.

5.      Similar to the previous major issue, so small is the cohort of patients reviewed that it is not possible or appropriate to present methodology about the administration of phage in a manner that could be read as advisory: e.g. 'how to administer phages'. Instead, I suggest presenting that information as 'experience of phage administration' or 'how phages were administered'.
Response: We agree and have modified accordingly.

6.      Line 149/150 - 'intravenous is not easy due to regulatory issues' - this statement is not justified and should be removed. I am not aware in my, or any other, context of any additional regulatory barriers to IV vs. local phage use.
Response: This sentence stems from the experience of one of the first clinical trials on phage therapy reporting that “the phage treatment was designed to be applied topically, avoiding a systemic injection, which was considered too risky by agencies in 2013 when we first proposed our study design. A systemic phage injection was potentially more prone to adverse events associated with endotoxin residues than a topical treatment would be, and so to meet with good manufacturing practices we designed a topical treatment”. However, we acknowledge that this is too less consistent to be reported and we followed the suggestion to remove it.

  1.      Line 157 - 'is not an alternative' - another generalisation. Phage therapy may be an alternative in some circumstances, e.g. allergy to antibiotics:https://www.ncbi.nlm.nih.gov/pmc/articles/PMC5494523
    Response: We re-phrased the statement and we thank the reviewer for the reference. The majority of phage experts now agree that phages are most impactful as an adjunct to antibiotics due to synergy. Phages as an alternative to antibiotics would be considered in cases where antibiotics could not be given at all, as in the case of allergies.

  2.      The discussion about phage/antibiotic interactions needs to be more succinct and precise. 'As showed in some studies in this review' is followed by no references. The discussion about phage/antibiotic interactions is not placed sufficiently in the context of broader evidence in this area.
    Response: We agree and have modified this section of the manuscript.

  3.      'Is there any pattern between management and outcomes?' This section doesn't really answer this question. Line 189 suggests answering this question is difficult due to the high risk of confounding factors, but these are not explored. There is no mention of this being challenging due to be extremely small number of patients considered in this review.
    Response: We added the limitations of the small numbers of subjects included.

    10.  The safety of phage therapy is such an important aspect of the evidence base that it requires greater consideration, both in scope and quality, than has been given. Line 200 speculates about all reported adverse events being related to the encounter between the phage preparation and the human immune system. This speculatory statement is not justified and should be amended or removed. Moreover, alternative and more plausible explanations are presented by the authors in the original reports of these adverse events. No detail is given on what a 'more standardised approach for the preparation of the phage treatment' means - presumably manufacturing in accordance with good manufacturing practice?
    Response: We agree and have removed the speculative statement.

  4.  It is unclear what the value the section about workflow adds. Suggest this is removed. The workflows will be different in different clinical settings and in different national contexts. As presented, the authors note that 'most' but not all cases were treated in specialised centres. The authors should avoid suggesting that only specialised centres are able to deliver phage therapy, as this is not the case as they have acknowledged. The authors also imply that the effort and regulatory requirements necessitate a specialised centre. It is incorrect to suggest that other centres are not capable of equally significant effort or are incapable of following the same regulations. It is also generalisation to state that collecting data and demonstrating the connectivity are often mandatory, when again the requirements here will vary markedly between national contexts.
    Response: We agree with this comment and we removed this section.

Minor issues

1.      There are more appropriate references regarding anti-biofilm activity than reference 8.

Response: We inserted a more appropriate reference (doi: 10.1128/AAC.01879-21).

2.      Rephrase line 48/49 - phages don't 'overcome' antibiotic resistance, as it's not a barrier phages need surmount.

Response: Thank you for the correction. We agree with that and modified the text accordingly.

3.      Software versions should be stated.

Response: This has been done.

4.      Line 87 - be consistent, use n=X/29 etc to orientate the reader.

Response: This has been corrected.

5.      Line 89 - 'combination of multiple' is tautologous.

Response: This has been corrected.

6.      Please reference the reports you refer to when discussing efficacy/safety.

Response: This has been done.

7.      Line 1030-104 typo 'wewewe' and duplication.

Response: This has been done.

8.      Ensure bacterial genus/species are presented consistently. I.e. in full first and then abbreviated, e.g. S. aureus. See line 138.

Response: This has been corrected.

9.      Line 143/144 does not read well.

Response: This has been corrected.

10.  Fig 2 has no labels; the legend suggests there should be.

Response: We apologize for the oversight. This has been corrected.

11.  There appear to be some instances of double spaces.

Response: This has been corrected.

Comments on the Quality of English Language
Requires moderate English editing. See comments for examples of areas requiring attention.

Reviewer 2 Report

Comments and Suggestions for Authors

The review «Phage Therapy for Cardiac Implantable Electronic Device and Vascular Graft: A Review of Clinical applications» presents the current state of the use of bacteriophages in clinical practice for the treatment of bacterial infections of cardiac implantable electronic devices. The authors performed a wide literature search in databases and found that to date only 14 studies have been published, in which a total of 29 patients participated. Undoubtedly, there is currently insufficient data for statistically significant conclusions about the effectiveness of the use of bacteriophages in this field of health care. However, it can be concluded that their further research is promising.

There may be a question about the need to separate the use of phages for the treatment of bacterial infections after implantation of cardiac implants into a separate topic. However, this problem differs from other phage therapy applications. «One of the main challenges   in treating CIED and vascular grafts infections is the presence of the biofilm matrix, which   acts as a protective shield, hindering the efficacy of antimicrobial agents and host immune   responses. Moreover, biofilm’s presence heightens the risk of systemic dissemination of   pathogens, complicating treatment and necessitating more aggressive interventions.»

The success of phage therapy depends on many different factors and it is often very difficult to compare the results obtained, especially for a small sample of patients. Despite this, the authors tried to answer a number of important questions, such as « How to select the patients who can potentially benefit from phage therapy? How to administer phages? and so on.

It follows from the manuscript that phage therapy is a promising complementary tool against CIED, requiring additional research.

Author Response

Response to Reviewer 2 Comments

We thank the reviewer for the thoughtful comments and offer our responses below. We feel that the manuscript is stronger as a result of the suggestions.

The review presents the current state of the use of bacteriophages in clinical practice for the treatment of bacterial infections of cardiac implantable electronic devices. The authors performed a wide literature search in databases and found that to date only 14 studies have been published, in which a total of 29 patients participated. Undoubtedly, there is currently insufficient data for statistically significant conclusions about the effectiveness of the use of bacteriophages in this field of health care. However, it can be concluded that their further research is promising.

There may be a question about the need to separate the use of phages for the treatment of bacterial infections after implantation of cardiac implants into a separate topic. However, this problem differs from other phage therapy applications.

The success of phage therapy depends on many different factors and it is often very difficult to compare the results obtained, especially for a small sample of patients. Despite this, the authors tried to answer a number of important questions, such as < How to select the patients who can potentially benefit from phage therapy? How to administer phages? and so on.

It follows from the manuscript that phage therapy is a promising complementary tool against CIED, requiring additional research.

Response: We thank the reviewer for the helpful and encouraging comments.

Reviewer 3 Report

Comments and Suggestions for Authors

Given the absence of reviews concerning phage therapy for cardiac implantable electronic devices and vascular grafts, this review is particularly intriguing. However, several points require attention from the authors:

1. Ensure the bacteria names are italicized throughout the text.

2. Verify the accuracy of the case numbers presented in Figure 1. For example, the total number of title screenings was reported as 77, with 66 exclusions. Consequently, the remaining papers should total 13.

3. Specify the number of replicates conducted in the methods section.

4. In line 150, the authors highlighted the rapid removal of phages by the human immune system. It is advisable for the authors to elaborate on the phage removal mechanisms in the review as well.

Comments on the Quality of English Language

The quality of the English language is good, but ensure the bacteria names are italicized throughout the text.

Author Response

Reviewer 3:

Given the absence of reviews concerning phage therapy for cardiac implantable electronic devices and vascular grafts, this review is particularly intriguing. However, several points require attention from the authors:

1. Ensure the bacteria names are italicized throughout the text.

Response: This has been corrected.

2. Verify the accuracy of the case numbers presented in Figure 1. For example, the total number of title screenings was reported as 77, with 66 exclusions. Consequently, the remaining papers should total 13.

Response: This has been corrected.

3. Specify the number of replicates conducted in the methods section.

Response: We did not report duplicates because we did not find any and because this is study is not being considered to be a systematic review.

4. In line 150, the authors highlighted the rapid removal of phages by the human immune system. It is advisable for the authors to elaborate on the phage removal mechanisms in the review as well.

Response: We felt that is exceeded the scope of this manuscript and have referenced comprehensive reviews on this topic.

Round 2

Reviewer 1 Report

Comments and Suggestions for Authors

Thank you for returning this manuscript, which is much improved.

Minor issues

·         Table 1 – please state IV in a consistent format

·         When referencing other authors, please use surname only, not initial. E.g. line 140 - Simpson et al. – see later in the text for Green and Blasco.

·         A few areas where English editing is required:

o   Line 132 – of whom 7 with…

o   Line 197 – bacteria biofilm

o   Line 212 – worth to be noted

o   Line 209 – supplementary enhancer = adjunct

o   Line 289 – insurgence

·         Tracked changes seems to suggest you no longer have a Figure 2 legend?

·         Line 239-240 – you should add a comment about the complexity of the phage/antibiotic interactions and that they are likely to be complex and vary by dose/phage/antibiotic

·         Line 293 – you mention good manufacturing. Do you mean GMP? For clarity, you may wish to use an alternative phrase such as high quality.

·         You may like to consider mentioning in your discussion that publication bias cannot be excluded from the review – i.e. how do you know more CIED patients haven’t been treated and not reported because phage therapy failed in all cases?

Comments on the Quality of English Language

See review comments.

Author Response

Reviewer 1 

Minor issues 

  1. Table 1 – please state IV in a consistent format. R: Modified. 
  2. When referencing other authors, please use surname only, not initial. E.g. line 140 - Simpson et al. – see later in the text for Green and Blasco. R: Modified.
  3. A few areas where English editing is required. R: We addressed all the requested English editing. 
  4. Tracked changes seems to suggest you no longer have a Figure 2 legend? R: We preferred not to report Figure 2 because the small diameter caused the catheter to become clotted and in retrospect, felt that this may be misleading to the audience.  
  5. Line 239-240 – you should add a comment about the complexity of the phage/antibiotic interactions and that they are likely to be complex and vary by dose/phage/antibiotic. R: we added a comment you can find in the modified text. 
  6. Line 293 – you mention good manufacturing. Do you mean GMP? For clarity, you may wish to use an alternative phrase such as high quality. R: Thank you.  It has been modified.
  7. You may like to consider mentioning in your discussion that publication bias cannot be excluded from the review – i.e. how do you know more CIED patients haven’t been treated and not reported because phage therapy failed in all cases? R: We agree with the reviewer that publication bias cannot be excluded and that this is important to note before interpreting the results. Therefore, we report a sentence at the beginning of the discussion. 

Reviewer 3 Report

Comments and Suggestions for Authors

The authors have improved the manuscript.

Author Response

Reviewer 2 

No suggestions to address.